# Photodimerization induced hierarchical and asymmetric iontronic micropatterns

Zehong Wang[1], Tiantian Li [1], Yixiang Chen[2], Jin Li[1], Xiaodong Ma[1], Jie Yin[1] & Xuesong Jiang [1] ✉

Micropatterning various ion-based modality materials offers compelling advantages for functionality enhancement in iontronic pressure sensing, piezoionic mechanoreception, and skin-interfaced electrode adhesion. However, most existing patterning techniques for iontronic materials suffer from low flexibility and limited modulation capability. Herein, we propose a facile and robust method to fabricate hierarchical and asymmetrical iontronic micropatterns (denoted as HAIMs) through programmed regulation of the internal stress distribution and the local ionic migration among an iontronic host. The resultant HAIMs with arbitrarily regulated morphologies and region-dependent ionic electrical performance can be readily made via localized photodimerization of an anthracene-functionalized ionic liquid copolymer (denoted as An-PIL) and subsequent vapor oxidative polymerization of 3,4-ethylenedioxythiophene (EDOT). Based on the piezoionic effect within the resultant distinct doped PEDOT, HAIMs can serve as a scalable iontronic potential generator. Successful syntheses of these fascinating micropatterns may accelerate the development of patterned iontronic materials in a flexible, programmable, and functionally adaptive form.

Hierarchical micro/nanopatterns enable biological surfaces to exhibit diverse properties and functions, such as antireflection, drag reduction, antiadhesion, power generation, and environment sensing, etc[1–4]. Among them, sensitive tactile perception is attractive and caused by collaboration systems between hierarchical microstructures and heterogeneous component molecules of the patterned biological surfaces[5,6]. For example, the sensation of the fingertip is due to (i) the conformal wrinkles on the fingerprint effectively transmitting an external stimulus to the subcutaneous receptors[7] and (ii) the established transmembrane potential caused by the asymmetric bilayer lipids inside and outside the cell membrane promoting stable and fast mechanotransduction[8,9]. Inspired by the bioelectronic mechanisms of biological surfaces, many artificial iontronic hosts have been developed and applied in spontaneous power generation[10], highly sensitive pressure sensors[11], electrochemiluminescence devices[12], and ionic heterojunction interfaces[13]. Nevertheless, most of them are prepared

via structural engineering techniques, i.e., photolithography, nanoimprinting, template transferring, and layer-by-layer deposition are utilized to construct hierarchical topographies (such as pyramids, cracks, wrinkles, and domes) or asymmetric iontronic interfaces[14–18]. Despite the fact that complicated surface patterns can be fabricated by top-down techniques with precise control[19], most of them involve lengthy and complex fabrication processes[20,21]. Moreover, the resultant patterns are fixed, difficult to dynamically regulate, and do not conform with the bioelectronic modalities that controls the flow, regions, and concentrations of ionic charge to interface with the environment[22]. More effective, easy-to-process, programming-controllable, and dynamically adaptable methods to manipulate ionic electrical performance for intronic materials are needed.

In contrast to traditional top-down approaches, biological surface patterns evolved from nature are usually spontaneous and self-organized, and generally involve the transfer of matter or energy and

[1]School of Chemistry & Chemical Engineering, Frontiers Science Center for Transformative Molecules, State Key Laboratory for Metal Matrix Composite Materials, Shanghai Jiao Tong University, 200240 Shanghai, China. [2]Key Laboratory of Science & Technology of Eco-Textile, Ministry of Education. College of Chemistry, Chemical Engineering and Biotechnology, Donghua University, 201620 Shanghai, China. ✉e-mail: ponygle@sjtu.edu.cn

interactions of compressive stress[23–26]. Among them, self-wrinkling is the most common pattern of living organisms and is regulated by compressive forces originating from gradients in stress normal to the surface or mismatches in properties between different layers of skins[27,28]. On this basis, self-wrinkling can be introduced onto the surfaces of iontronic materials, and then the internal stress distribution can be varied with an external stimulus to simultaneously achieve dynamic regulation of the surface topography and internal ionic charge distribution, thereby establishing a hierarchical and heterogeneous iontronic surface. Photoinduced cross-linkable poly (ionic liquids; denoted as PILs), as the forefront of advanced and versatile iontronic materials, have high ionic conductivity, good processability, substantial amounts of mobile cations and anions, and high compatibility with small polar compounds[29,30]. These materials would be, in this regard, exceptional candidates for constructing hierarchical iontronic patterns. However, since the mobile ions in homogeneous PILs can only migrate and redistribute in small regions along with the polymer chains before and after photocrosslinking, it is difficult to dramatically vary the ionic electrical performance for the surface patterns to fit the need. Alternatively, developing a heterogeneous PIL system that can synchronously control the morphologies of iontronic patterns and greatly change the ion concentration along with different cross-linked regions would be more promising.

According to our previous work, surface wrinkling can dynamically and precisely regulate and arbitrarily tune the topographies of surfaces in two dimensions[31]. Herein, we present a robust method with which to create hierarchical and asymmetric iontronic micropatterns (denoted HAIMs) by incorporating masked photodimerization and asymmetric vapor oxidative polymerization supported by an in-situ phase separation process. A random copolymer comprised of two groups (anthracene-functionalized ionic liquid and n-butyl acrylate, denoted as An-PIL) with distinct solubility in 1-ethyl-3-methylimidazolium bis(trifluoromethylsulfonyl)imide (denoted as IL) was prepared to accumulate internal stress upon UV exposure and readily regulate the stress distribution to produce self-wrinkling patterns under a mask. Moreover, an inhomogeneous iontronic host comprising of An-PIL and IL was developed to realize the goal of in-situ phase separation and thus vary the mobile ionic species among different patterned regions and also support the vapor oxidative polymerization. The strategy combines the advantages of a top-down method (precise and controllable), self-organized wrinkling (simple and natural), and bioelectronics mechanisms (tunable ionic charge and action potential), paving the way for simple, controllable, and operable manufacturing of organic electronics. The resultant HAIM exhibits the integrated properties of a highly polarizable PIL, ease of mobile ionic charge, capacitive and potentiometric gradient controllability, and long-term durability. The incorporation of photodimerization to fabricate HAIM was demonstrated to achieve promising bioelectronic modalities either on macro or micro scales.

## Results

### Syntheses and ionic electrical modalities of HAIMs

We prepared HAIMs satisfying three criteria: (i) the iontronic host can effectively crosslink via anthracene group under UV exposure, (ii) the iontronic host must be able to store ionic charges and readily redistribute them under external stimulation, and (iii) the iontronic host must be further processable to construct hierarchical and asymmetric patterns through controllable gas phase polymerization. The first two requirements were satisfied by developing an An-PIL/IL mixture with minimal free energy, among which An-PIL was used as photodimerization functional matrix to generate micropatterns and microstructures[28], and a liquid IL with a substantial amount of mobile ionic charge acted as a migrant phase to realize ionic redistribution. To satisfy the last criterion, vapor oxidative polymerization of 3,4-ethylenedioxythiophene (EDOT) was carried out in a nonuniform chemical

environment, and the resultant hierarchical morphology and asymmetrical iontronic performance came from the two rounds of internal stress regulation as well as the distinct IL-doped PEDOT synthesized upon exposure and previously unexposed regions.

Figure 1a describes the overall chemical reactions and synthesized molecular structures. A random copolymer, An-PIL, was synthesized to bring about photocrosslinking for the iontronic host. Moreover, the IL was used as a plasticizer and a separable moiety to vary the glass transition temperature ($T_g$) of the An-PIL and provide a microreactor for the synthesis of IL-doped PEDOT. The synthesis and characteristics of An-PIL are provided in Supplementary Figs. 1 and 2. The fabrication process began with preparation of the An-PIL/IL film (i.e., iontronic host) comprising different molar ratios of IL to the anthracene units of An-PIL. Subsequently, the iontronic host became highly miscible and well dispersed with the help of the similar structural and closed characteristics of the IL and An-PIL backbones (i.e., the solubility parameter close principle[32]). Due to the absorption attenuation of UV light normal to the surface of the iontronic host, a gradient occurred for photodimerization, which induced compressive stresses between the rigid crosslinked surface and the underlying pristine An-PIL. When the incident UV light was masked with a pattern, a compressive force was only generated and distributed across the exposed regions, allowing in-plane and out-plane stress relaxation in the exposed surface and giving rise to patterning or buckling while the unexposed regions remained flat[28]. Therefore, an ordered surface pattern replicated the mask produced on the surface of the iontronic host. Primary patterns can readily be generated by localized photodimerization of the iontronic host with photomasks, which can be understood as self-wrinkling appearing in the exposed region due to gradient photocrosslinking (Fig. 1b). In addition, in-situ phase separation occurred over the exposed regions after heating at 85 °C for 15 min, while the IL droplets were secreted from the iontronic host and firmly adhered to the primary patterns. Notably, IL droplet secretion only occured in the exposed region (i.e., primary patterns) and can be washed out by ethanol, as shown in Supplementary Fig. 3.

To further build hierarchical micropatterns onto the iontronic host, vapor oxidative polymerization of EDOT was conducted, resulting in secondary wrinkle formation on previously unexposed regions. Moreover, the IL droplets on primary patterns acted as microreactors to produce IL-doped PEDOT, while pure PEDOT was synthesized on the surfaces of secondary wrinkles. Ultimately, formation of asymmetrical iontronic micropatterns. Figure 1c shows an optical image of a typical HAIM with a hierarchical surface containing stripe patterns (i.e., primary patterns) and random wrinkles (secondary wrinkles). A representative field-emission scanning electron microscopy (FE-SEM) image of the relevant surface morphology is presented in Fig. 1d, e revealed that spherical IL-doped PEDOT particles with diameters of ~5 μm aggregated onto primary patterns, while flaky pure PEDOT was widely distributed onto the secondary winkles.

Figure 1f demonstrates the proposed mechanisms for the ionic migration behavior of HAIMs in response to external compression. Owing to the presence of mobile ionic charges in IL-doped PEDOT and once HAIMs were squeezed, the mobile ionic charges are depolarized and compelled to spatially redistribute within the scaffolding polymer matrix, which we refer to as the piezoionic effect[33–35]. Moreover, there is a difference between the transport capabilities of the ionic charges in the PEDOT/IL-rich domain in response to the built stress distribution, i.e., the cations in the IL are positively charged, and the interactions between cations and the main chain of PEDOT are weaker; thus, it is easier for cations to move to the opposite side away from the external force. Meanwhile, due to the stronger electrostatic forces between anions ($I^-$ and $TFSI^-$) and the positively charged PEDOT, the anions remain relatively near the surface. Ultimately, a negative space charge near the interface of the external force and a positive space charge at the interface far from the external force were generated. In

addition, since there were no excess mobile ionic charges in pure PEDOT on secondary wrinkles, it will be electrically neutral under an external force. Therefore, based on the distinct piezoionic effects of hierarchical iontronic micropatterns, the resultant HAIM exhibits asymmetrical ionic electrical performance and a potential gradient under the action of an external force when connected to an external circuit.

**Photodimerization and phase separation of an iontronic host**
To clarify the mechanism of photodimerization inducing subsequent phase separation of iontronic host, we conducted a series of controlled experiments and used the corresponding IL molar ratio-dependent micropattern morphology variations to determine the necessary factors for the initial photocrosslinking event and the resultant IL droplet secretion. The iontronic host comprising of different molar ratios of An-PIL to IL was exposed to UV light with a photomask to achieve local photodimerization and then heated to achieve secretion of IL droplets onto the primary patterns (Fig. 2a–c). As the molar ratio of An-PIL to IL increased, the amplitude (i.e., the height) of the resultant primary patterns grew from 0 μm (the molar ratio of An-PIL to IL was 1:0, i.e., the corresponding molar content of IL was 0%) to 20 μm (the molar ratio of An-PIL to IL was 1:10, i.e., the corresponding molar content of IL was 91%), as shown in Fig. 2d. Subsequently, after heating the obtained primary patterns at 85 °C for 15 min, IL droplets began to secreted from the surfaces of the primary patterns when the molar ratio of An-PIL to

IL was 1:1 (i.e., the corresponding molar content of IL was 50%, as shown in Supplementary Fig. 4). Moreover, the diameters of the IL droplets were reduced from 19.7 μm to 12.9 μm until they finally disappeared as the molar ratios of IL further increased. Remarkably, the as-prepared primary pattern (corresponding to the exposure region) has a clear boundary (Fig. 2b), and the IL droplets are confined to the boundary of the primary pattern.

To gain insight into the mechanism for IL droplet secretion, we investigated the laser scanning confocal microscopy (LSCM) image and the corresponding phase images of a representative primary pattern in Fig. 2b. To better illustrate the internal phase structure changes of the iontronic host, we divided the resultant surface patterns into three regions: region I represents the unexposed region (i.e., the pristine iontronic host), region II represents the transition region between the unexposed region and exposed region, and region III represents the exposed region (i.e., IL droplets @primary patterns), as shown in Fig. 2e, f. The LSCM image in Fig. 2f shows that evenly spaced stripe patterns were covered by IL droplets, whereas the transition region and unexposed region were flat and had no obvious change. In addition, the phase image of the unexposed region (Fig. 2g) shows a distinct nanophase-separated structure in region I. Based on the microphase separation occurring in the unexposed region and according to the "like dissolves like" rule, we propose that the weak polarities of BA segments drive aggregation to form phase-separated BA-rich domains. In contrast, the strong polarity of An-IL enables the

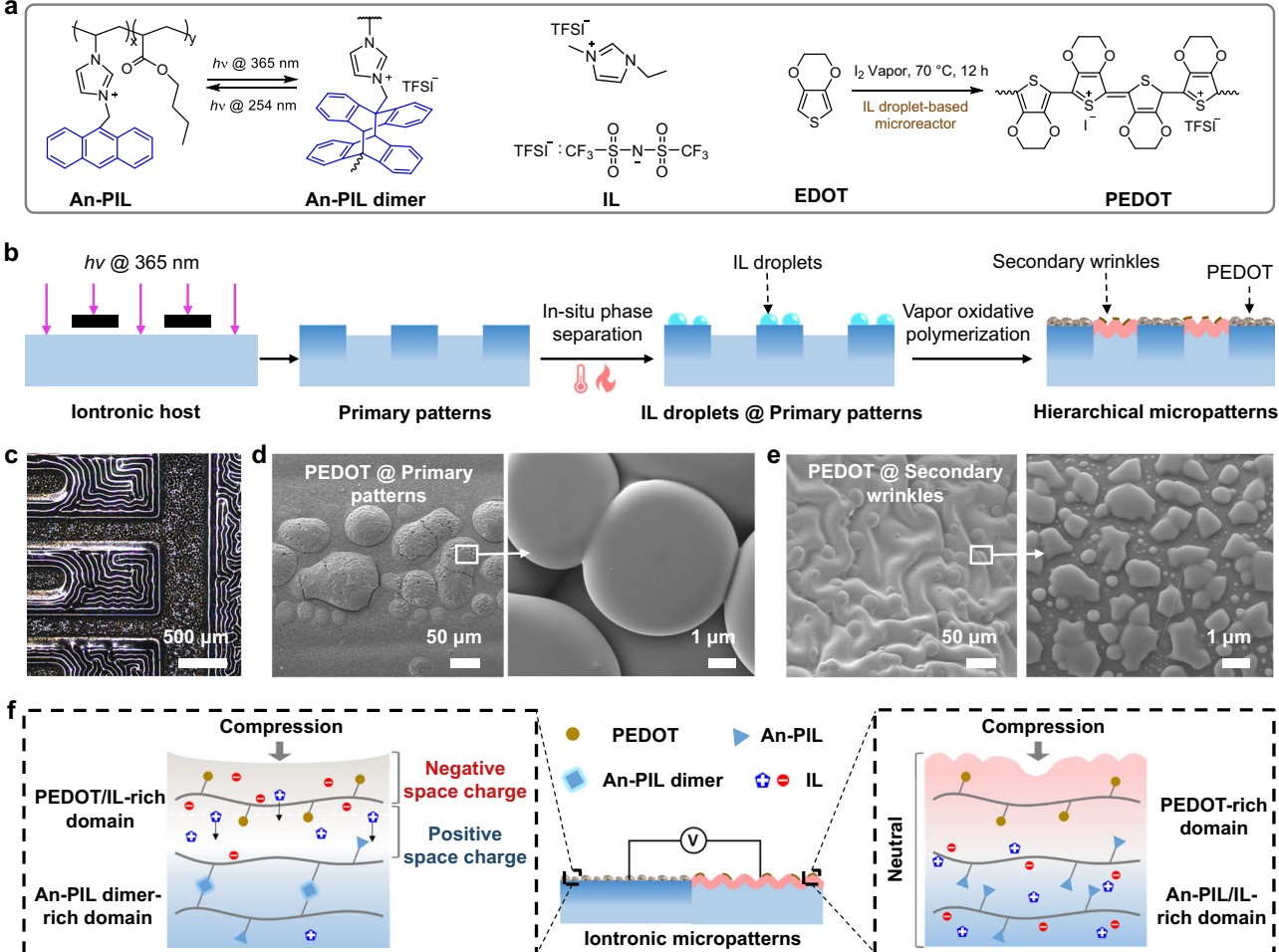

**Fig. 1 | Design, structure, and piezoionic effect of a HAIM. a** Chemical structure of An-PIL, An-PIL dimer, IL, and PEDOT. **b** Schematic of the fabrication process for HAIMs through localized photodimerization, heating (to drive in-situ phase separation), and vapor oxidative polymerization. **c** Microscope images of a representative HAIM sample. FSEM images of PEDOT on **d** primary patterns and **e** secondary wrinkles. **f** Proposed mechanism for formation of the potential gradient between IL-doped PEDOT @primary patterns and pure PEDOT @secondary wrinkles.

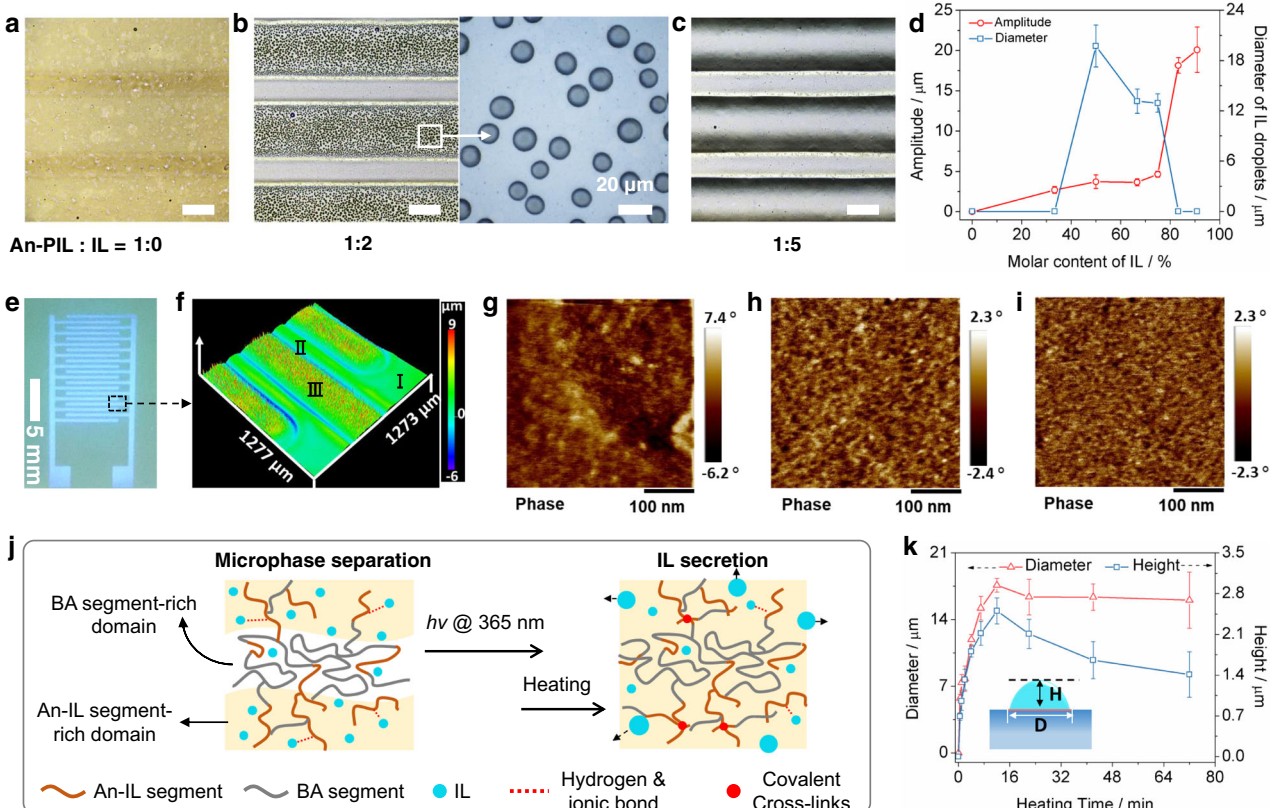

**Fig. 2 | Photodimerization and in-situ phase separation of the iontronic host.** **a–c** Microscopy images of iontronic hosts after photodimerization crosslinking and heating; scale bar: 200 μm. **d** Amplitude variations of the primary patterns and diameter changes of the IL droplets versus the molar content of IL. Error bars represent the standard deviations of three independent data. **e** Fluorescence image and **f** the corresponding LSCM image of the IL droplets @primary patterns; I, II, and III represent the unexposed region, transition region, and the exposed region, respectively. AFM phase images of the **g** unexposed region, **h** transition region, and **i** exposed region corresponding to **f**. **j** Proposed phase separation mechanisms for iontronic hosts after photodimerization crosslinking and heating. **k** Diameter (D) and height change (H) of the IL droplets versus the heating time, the inset represents the wetting model of IL droplets on primary patterns. Error bars represent the standard deviations of three independent data.

An-IL-rich domains to be surrounded by IL, as shown in the pristine iontronic host (Fig. 2j). Furthermore, the brighter region in the phase image (Fig. 2g) corresponds to the hard region (i.e., the An-IL moiety), and the darker region corresponds to the soft segment (i.e., the BA unit). After localized photodimerization and heating to secrete IL droplets, a relatively uniform miscible phase structure was formed in region III (Fig. 2i). Moreover, although the transition region (II) was unexposed, the phase image distribution (Fig. 2h) appears more uniform than that of the pristine unexposed region (I). Therefore, photodimerization enables more An-PIL dimers to be generated, and more IL becomes freely mobile in the primary patterns. After heating, the IL migrates and coalesces and is finally secreted out within the surfaces of the primary patterns, resulting in a more homogenous phase distribution (Fig. 2i). Moreover, the mobile IL may migrate from the transition region to the exposed region and thus enable the An-IL and BA segment units in region II to become more homogenous.

To investigate the role of heating on IL phase separation, we heated the photodimerization cross-linked patterns (among which the molar ratios of An-PIL to IL were 1:2) at 85 °C and analyzed the diameter variations of IL droplets as a function of heating time, as shown in Fig. 2k. As the heating time was increased, the diameters and heights of the IL droplets increased from 0 μm (0 min) to 17.6 μm and 2.5 μm (12 min), respectively, and then slightly decreased until remaining unchanged. Based on the above results, we propose that the initial mixture of the An-PIL dimer and IL is thermodynamically unstable, and the IL tends to phase separate and readily aggregate into larger droplets after heating. Moreover, since An-IL segments have good

compatibility with IL due to the ionization of the imidazolium group and hydrogen bonding interaction, a miscible and stable iontronic host could be formed initially (Supplementary Fig. 5). However, due to the localized photodimerization, the amount of An-IL that originally interacted with ILs in the exposed region decreased, and therefore, the IL droplets freely moved in the system. As the heating time was increased, the free IL droplets began to collide, coalesce and grow and were finally secreted out of the surfaces of the primary patterns. Regarding the nonexposed region, the interaction between IL and An-IL was unchanged, and thus no IL droplet separation occurred. In addition, when the molar content of IL was too high, An-PIL tended to form a highly solvated network structure in the IL bulk, and both An-IL and BA segments were solvated by large amounts of IL; thus, the iontronic host became highly soft, and the free IL droplets were more likely to move into the IL bulk instead of being secreted.

## Ionic electrical performance of IL droplets @primary patterns

Due to the flexibility and adaptability of the photodimerization crosslinking and heat treatment process, a series of IL droplets @patterns (where the molar ratio of An-PIL to IL in the iontronic host was 1:2) were readily fabricated, as shown in Fig. 3a. Due to the localized secretion of IL, which resulting in different contents of mobile anions and cations within the unexposed and exposure regions, various micropatterns with distinct ionic conductivities can be fabricated by using the corresponding photomasks and simultaneously controlling the subsequent heating time. To illustrate the unique iontronic features of IL droplets @primary patterns,

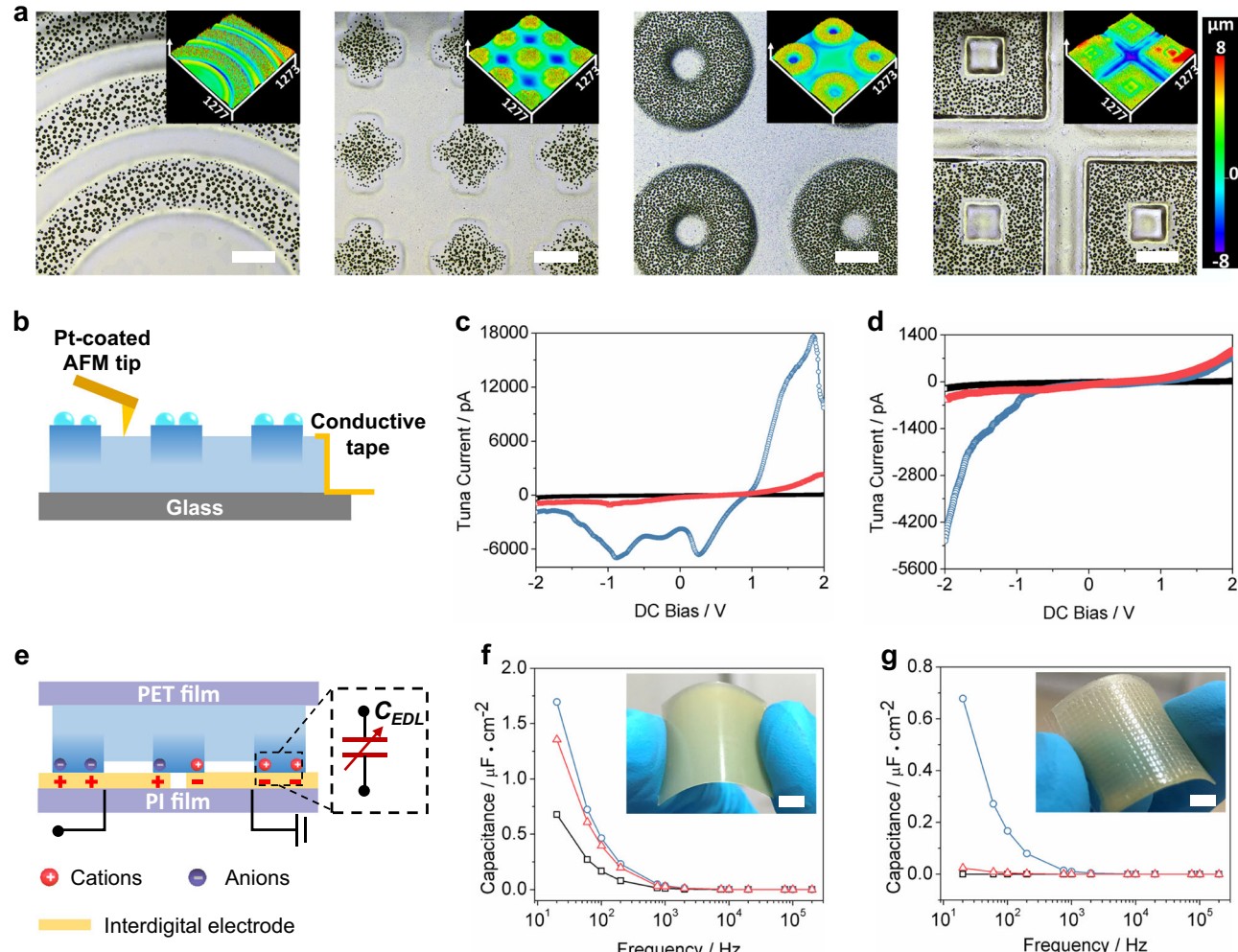

**Fig. 3 | Localized regulation of the ionic electrical performance of primary patterns. a** Microscopy images showing IL droplets @ primary patterns, scale bar: 200 μm. **b** Schematic diagram of the conductivity AFM measurement (C-AFM) for IL droplets @primary patterns. Averaged point contact I-V curve obtained after averaging I-V responses from 15 different spots shown for **c** unexposed regions (i.e., pristine iontronic host) and **d** exposed regions (i.e., IL droplets @primary patterns) with molar ratios of An-PIL to IL were 1:1 (black line), 1:2 (gray-blue line), and 1:3 (red line), respectively. **e** Schematic diagram of the EDL capacitance between interdigital electrodes and IL droplets @primary patterns. Interfacial capacitance changes at different frequencies for **f** unexposed regions (i.e., pristine iontronic host) and **g** exposure regions (i.e., IL droplets @primary patterns) with different molar ratios of An-PIL to IL were 1:1 (black line), 1:2 (gray-blue line), and 1:3 (red line), respectively; the inset shows the corresponding optical images of the pristine iontronic host and micropatterned iontronic host with molar ratios of An-PIL to IL were 1:2, scale bar: 1 cm.

conductive AFM (C-AFM) measurements (Fig. 3b) were used to determine the point contact I-V curves for different regions corresponding to representative doughnut-like primary patterns. As shown in Fig. 3c, d, the positive and negative currents exhibited obvious nonlinear characteristics for both unexposed and exposed regions, indicating that a Schottky contact was formed between the Pt-coated tips and primary patterns. In addition, the positive bias current was significantly higher than that of the negative bias current for the unexposed region (i.e., the pristine iontronic host), as shown by comparing the current changes in Fig. 3c. According to the ionic conduction mechanism[36–38], ion transport among ionic polymeric networks is driven by ion jumps over an energy barrier among coordinative polar groups on the polymer matrix. For the iontronic host in this work, the $T_g$ decreased with the addition of more IL (Supplementary Fig. 6), and thus, the free ions were easily transported among An-PIL. Moreover, the anions (free [TFSI]⁻ groups) were bound to the polyimidazolium cations of An-PIL via electrostatic forces. Thus, the cations (free imidazolium groups) were the main mobile ions and resulted in a rectification effect under positive and negative bias. In addition, due to the cross-linked polymeric ionic network generated

in the exposure regions, free ion transport was difficult, and thus, the current for the exposed region was smaller than that of the unexposed region, as shown in Fig. 3c, d. Further results on the current curves showed that the tunnel currents for both unexposed and exposure regions were associated with IL content. The largest tunnel current appeared with an An-PIL to IL molar ratio of 1:2, which can be explained by the intrinsic phase structure change of the iontronic host. When less IL was added to An-PIL (i.e., the molar ratio of An-PIL to IL was 1:1), the rigid polymeric ionic network hindered the ionic conduction and fewer free ions could be transported above the $T_g$ of the iontronic host at the same bias voltage, resulting in a smaller tunneling current. In comparison, when excess IL was added into An-PIL (i.e., the molar ratios of An-PIL to IL were 1:3), the iontronic host became softer, and a substantial amount of IL accumulated at the bottom layer of the iontronic host because the density of IL was higher than that of An-PIL. Therefore, the free ion content at the interface between the pattern and the tips was lower, and this resulted in a smaller tunnel current. Thus, the ionic conductivities of the patterns can be regulated by controlling the molar ratio of An-PIL to IL in the iontronic hosts.

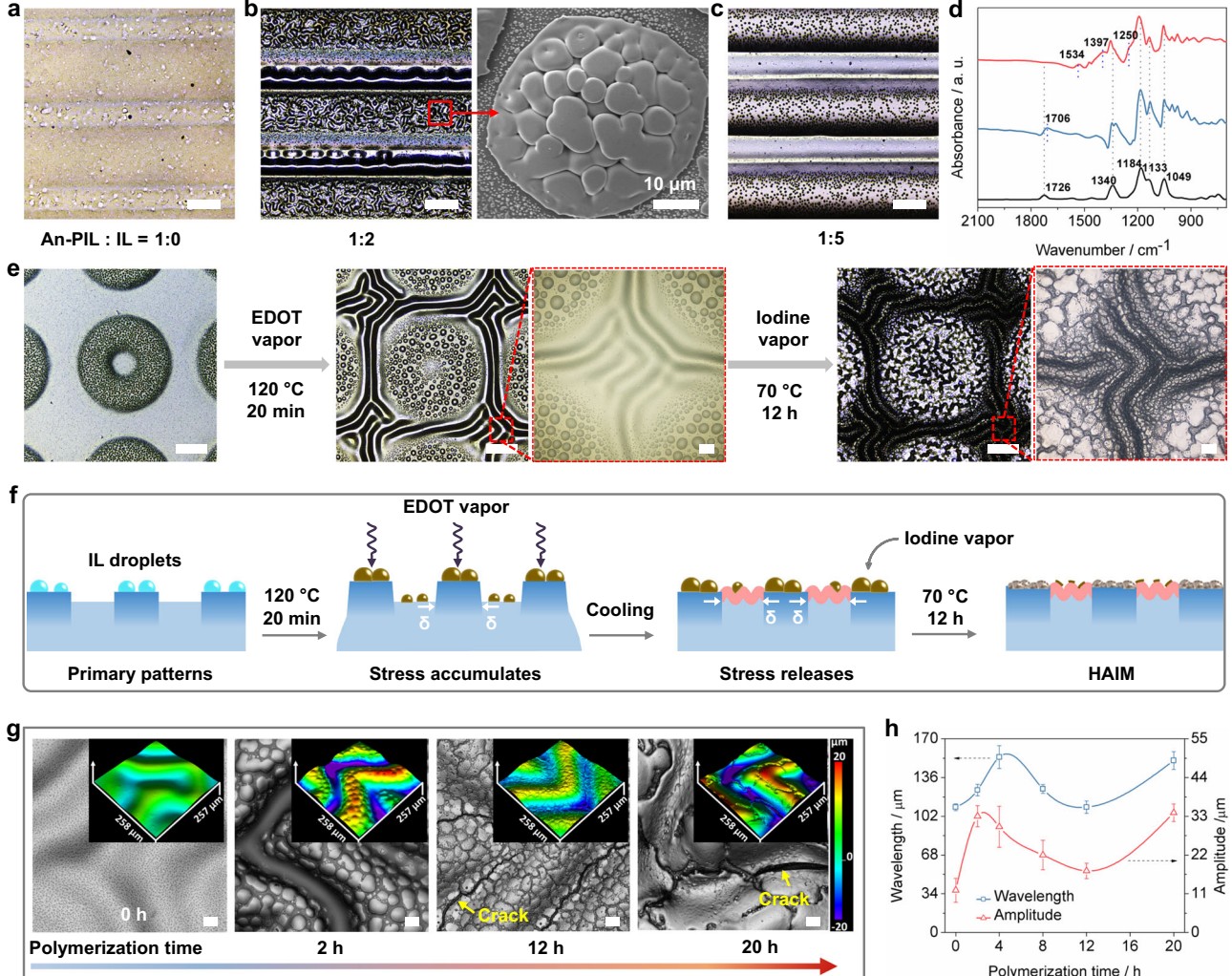

**Fig. 4 | Fabrication, characterization, and mechanisms of secondary wrinkles formation. a–c** Optical microscopy images of PEDOT synthesized on the surface of primary patterns containing different molar ratios of An-PIL to IL, scale bar: 200 μm. **d** FTIR spectra for the pristine iontronic host (black line), IL droplets @primary patterns (gray-blue line), and the resultant IL-doped PEDOT @primary patterns (red line). **e** Optical microscopy images of the morphology variations for fabrication of the secondary wrinkles, scale bar: 200 μm, scale bar of enlarged images: 20 μm. **f** Schematic demonstration of the fabrication of HAIMs with hierarchical and asymmetrical morphologies. **g** Optical microscopy images of secondary wrinkles as a function of polymerization times; scale bar: 50 μm, the inset was the corresponding LSCM images. **h** Wavelength and amplitude change of secondary wrinkles versus polymerization time. Error bars represent the standard deviations of three independent data.

In addition, the interfacial capacitance of iontronic hosts before and after micropatterning was investigated. Based on the polarization characteristics of An-PIL and the large number of free ions in the iontronic host, an electric double layer (EDL) capacitance can readily be generated between the accumulated positive and negative charges on the two opposite electrodes and the corresponding cations and anions, as shown in Fig. 3e. The resultant interfacial capacitance as a function of excitation frequency and molar ratios is summarized in Fig. 2f, g. As expected, the EDL capacitance was highly frequency dependent, and the unit-area capacitance gradually decreased with increasing frequency. The capacitance-frequency curves of the exposed samples (i.e., the IL @primary patterns) exhibited a trend similar to those of unexposed samples (i.e., the pristine iontronic host) over the same frequency range (20 Hz to 2 MHz). In particular, the unit-area capacitances of the pristine iontronic hosts were 1.69 μF cm$^{-2}$, 1.35 μF cm$^{-2}$, and 0.67 μF cm$^{-2}$ (20 Hz), for An-PIL to IL molar ratios of 1:2, 1:3, and 1:1, respectively, as shown in Fig. 3f. However, for the IL @primary patterns containing different molar ratios of An-PIL to IL, the unit-area capacitances declined to 0.68 μF cm$^{-2}$, 0.025 μF cm$^{-2}$, and 0.002 μF cm$^{-2}$, respectively, as shown in Fig. 3g. Thus, higher molar ratios of IL in

iontronic hosts result in appreciable declines in the interfacial capacitance at the same excitation frequency. Finally, with the requirements of higher capacitance and a highly patterned surface, the IL @primary patterns composed of 1:2 molar ratios of An-PIL to IL were the optimal samples for subsequent construction of secondary wrinkles.

## Construction of secondary iontronic wrinkles

Owing to the high solubility of 3,4-ethylenedioxythiophene (EDOT) in [EMIm][TFSI] (i.e., the IL used in this work), the IL @primary patterns can be developed as microreactor vessels for the synthesis of PEDOT, as shown in Supplementary Fig. 7. The resultant PEDOT deposited on the primary patterns is shown in Fig. 4a–c. The IL droplets first absorbed EDOT vapor and then processed it with oxidative polymerization in the presence of iodine vapor. Thereafter, IL-doped PEDOT was synthesized, and secondary wrinkles appeared simultaneously, as shown in Fig. 4b. Notably, when the primary patterns contained a lower or higher molar ratio of IL (i.e., relative to the molar ratios of An-PIL to IL in Fig. 4a, c, which were 1:1 and 1:5, respectively), the IL droplets will not be secreted onto primary patterns as in Fig. 2, and therefore, the EDOT vapor was rarely absorbed on the surfaces of

primary patterns; ultimately no PEDOT was generated in Fig. 4a and a few randomly distributed grainy PEDOT appeared in Fig. 4c. Subsequently, microscopic attenuated total reflection infrared spectroscopy (ATR) was used to characterize the surface component variations of IL droplets @primary patterns after vapor oxidative polymerization of PEDOT (Fig. 4d). The characteristic peaks at 1726 cm$^{-1}$ were assigned to ester groups of the BA segments on An-PIL, and bands located at 1340 and 1184 cm$^{-1}$ were attributed to asymmetric and symmetric stretching vibrations of the O=S=O groups of TFSI$^-$, respectively. In addition, the peak at 1049 cm$^{-1}$ was the characteristic stretching vibration of S-N-S moieties in TFSI anions[39,40]. Since the chemical structure of the IL was similar to that of An-PIL, there were no significant changes in the characteristic peaks after photodimerization and heating of the iontronic host, as shown in the spectra of IL @primary patterns. Moreover, new peaks at 1534, 1397, and 1250 cm$^{-1}$ were attributed to the C=C and C-C stretching vibrations of the quinoid structure of the thiophene ring, respectively. In addition, the band at approximately 1250 cm$^{-1}$ was due to C–O–C bond stretching in the ethylene dioxy (alkylene dioxy) group. This series of bands was consistent with the literature[41,42], suggesting formation of the IL-doped PEDOT.

To understand the buckling mechanism for secondary wrinkles, we tracked the morphological changes of the primary patterns, as shown in Fig. 4e. As EDOT evaporated and was adsorbed into IL droplets at 120 °C, secondary wrinkles were generated in the unexposed regions of the primary patterns after cooling. Additionally, the diameters of the IL droplets became larger, and the primary patterns swelled after absorbing EDOT due to the morphology change shown in Supplementary Fig. 8a–d. After further processing under an atmosphere of iodine vapor, IL-doped PEDOT was synthesized on the surfaces of the primary patterns, while pure PEDOT (i.e., non-IL doped PEDOT) was generated on the secondary wrinkles. To clarify the reasons for production of secondary wrinkles, a modified bilayer system that incorporated stiff primary patterns (surface layer) and a soft iontronic host (substrate) was considered in Fig. 4f based on our previous work[28,43]. Since the internal polymeric matrix of the primary pattern was crosslinked and exhibited a higher modulus, the pristine iontronic host (substrate layer) had a larger expansion coefficient relative to those of the primary patterns. Therefore, thermal expansion of the unexposed areas near the primary patterns was bound by the finite boundary during heating treatment, resulting in accumulation of internal stress at the interface between the primary patterns and unexposed regions. Moreover, EDOT vapor was more inclined to be adsorbed by IL droplets, and small amounts of EDOT vapor were also adsorbed onto unexposed regions. Therefore, the internal stress accumulated in the stiff primary patterns was released during cooling, and the primary patterns became flat and expanded (Supplementary Fig. 8d–f) while squeezing the softer unexposed regions to produce the typical wrinkle pattern (Supplementary Fig. 8g–i). At the same time, IL-doped and raw PEDOT were produced on the surfaces of primary patterns and secondary wrinkles, respectively, due to oxidation by the iodine vapor.

In addition, by controlling the polymerization time of PEDOT, secondary wrinkles with different amplitudes can be regulated as shown in Fig. 4g. This process can also be illustrated as a typical bilayer model with secondary wrinkles acting as a softer substrate layer and the resultant rigid PEDOT serving as the stiff surface layer. When the vapor polymerization time was shorter than 2 h, the peaks of the secondary wrinkles was first exposed to iodine vapor, and thus, PEDOT was first generated on the peaks, which resulted in increased stress and thickness of the rigid surface layer over the peaks and finally caused the wavelength and amplitude to increase sharply[44]. When the reaction time was longer than 4 h, PEDOT was gradually produced in the valleys of the secondary wrinkles, and therefore, the heights of the valley also increased. At this point, the height difference between the peaks and the valleys (i.e., the amplitude) decreased, and therefore, an obvious

decline appeared in the amplitude in Fig. 4h. Moreover, with thicker raw PEDOT produced on the peaks and valleys of secondary wrinkles when the polymerization time was extended to 12 h, dense raw PEDOT and cracks were formed among the resultant PEDOT @secondary wrinkles because the excess stress concentration within bilayers comprising of a higher modulus of raw PEDOT (1.2 GPa in Supplementary Fig. 9) and pristine soft secondary wrinkles. The amplitude and wavelength of the secondary wrinkles dropped sharply owing to the dramatic release of stress by crack formation. Moreover, when the reaction time was further increased to 20 h, the resultant raw PEDOT layers were further built up and covered on the peaks, valleys, and cracks, and many isolated "smooth wrinkles" appeared on the secondary wrinkles because additionally compressive stress accumulated in the wavelength and amplitude directions; and thus, the wavelength and amplitude also increased. Therefore, considering the structural integrity and electrical conductivity of PEDOT, HAIM prepared at 12 h will be used for the following study.

### Asymmetric ionic electrical performance of HAIMs

Figure 5a, b shows the morphology of the as-prepared HAIMs. Owing to the boundary constraints of the primary patterns, long-range wrinkle patterns were produced in area I parallel to the boundary on the surfaces of the unexposed region. However, labyrinth-like wrinkle patterns were generated in the unexposed regions in area II. The distinct wrinkling patterns were caused by the different boundary constraint conditions[31]. The LSCM images in Fig. 5c show that the amplitude of labyrinth wrinkles in area II was lower than that of the parallel wrinkles in area I, which was consistent with the results in Fig. 4g, h. In addition, Fig. 5d shows spherical PEDOT particles bonded to each other in area I, while flake PEDOT was distributed discretely in area II. In terms of area I (i.e., IL droplets @primary patterns), polymerization of EDOT was carried out in the IL droplets. Moreover, due to the strong polarity of the IL and the electrostatic interactions between the IL and the PEDOT main chain, it was difficult to form a linear arrangement and stacking for the resultant spherical IL-doped PEDOT[45]. However, in the chemical environment of area II, the EDOT was polymerized on the unexposed regions and without IL droplet doping; therefore, the structure of the as-prepared PEDOT tended to be complete and flaky. To further illustrate the asymmetric shape of the resultant PEDOT, energy dispersive spectroscopy (EDS) tests were conducted for PEDOT on both primary patterns and secondary wrinkles, as shown in Fig. 5e and Supplementary Fig. 10. Comparing the changes in the surface atomic weight percentage for different regions showed that the nitrogen content in the PEDOT @secondary patterns (area II) was close to 0%, while that in the PEDOT @primary patterns (area I) was 3.85%. Moreover, since the EDOT itself does not contain nitrogen, the nitrogen in polymeric EDOT can only come from the IL doping. In addition, since iodide ions were the counterions for pure PEDOT in area II while [TFSI]$^-$ groups may also become the counterions of IL-doped PEDOT in area I, the iodide content in area II (22.54 %) was obviously higher than that of area I (11.53%). In summary, the above results showed that asymmetric PEDOT was synthesized on primary patterns by developing different vapor polymerization environments.

The ionic electrical performance of the as-prepared HAIM was determined with homemade encapsulation devices, as shown in Fig. 5f. The interfacial capacitance (Fig. 5g) was obtained by connecting point B and point C to a digital multimeter. Since IL secretion and thus the number of mobile ions were reduced in primary patterns, the capacitance decreased from 1.69 μF cm$^{-2}$ for the pristine iontronic host to 0.68 μF cm$^{-2}$ for the IL @primary patterns and to 0.007 μF cm$^{-2}$ for HAIM at the same frequency of 20 Hz. In addition, the surface resistivity (Fig. 5h) was determined by connecting point A and point B or C to a digital multimeter. With the production of conductive PEDOT on the surface of the iontronic host, its surface resistivity decreased from the original 3.7 MΩ•cm$^{-1}$ to 2.33 MΩ•cm$^{-1}$ for the PEDOT @exposed

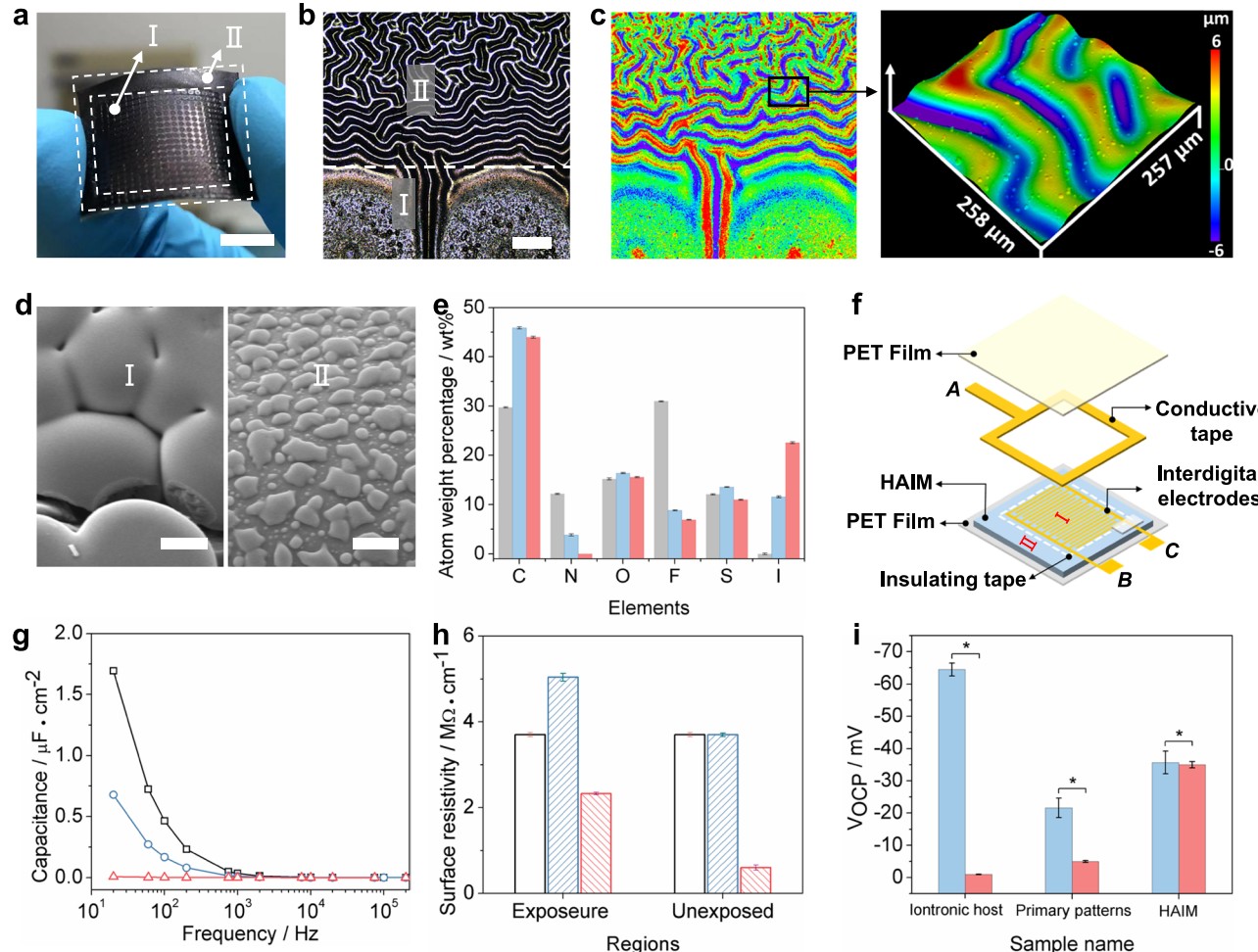

**Fig. 5 | Characterization of the ionic electrical performance of the HAIM.**
**a** Optical images of the as-prepared HAIM, scale bar: 1 cm; I and II represent the produced PEDOT on exposed and unexposed areas, respectively. **b**, **c** Microcopy and LSCM images of the HAIM corresponding to **a**, scale bar: 200 μm. **d** SEM images of the resultant PEDOT in different areas corresponding to **b**, scale bar: 2 μm. **e** Atom weight percentage changes of pristine iontronic host (gray column), IL-doped PEDOT @primary patterns (gray-blue column), and raw PEDOT @secondary wrinkles (red column), respectively, were obtained by EDS mapping. **f** Homemade devices for electrical measurements of different samples in this work.

**g** Capacitance changes versus frequency for pristine iontronic host (black line), IL-doped PEDOT @primary patterns (gray-blue line), and resultant HAIM (red line), respectively. **h** Surface resistivity of different regions in pristine iontronic host (black frame), IL-doped PEDOT @primary patterns (gray-blue frame), and resultant HAIM (red frame), respectively. Error bars represent the standard deviations of five independent data. **i** Potential gradients changes of different samples within a limited time interval (gray-blue column, red column, and asterisk were initial potential, final potential, and 1 min interval, respectively). Error bars represent the standard deviations of three independent data.

regions (area I) and to 0.6 MΩ•cm⁻¹ for the PEDOT @unexposed region (area II). Thereafter, the asymmetric conductive micropatterns provided the necessary conditions for realization of the piezoionic effect and resulted in a potential gradient between area I and area II, as shown in Fig. 5i. Generally, the piezoionic effect is a result of Donnan-like depolarization due to an inhomogeneous ionic distribution[33]. In terms of the HAIM, the IL-doped PEDOT initially at equilibrium had a uniform distribution of mobile species, such that the electrochemical potentials experienced by all species were equal and the free energy of the system was minimal. When a mechanical perturbation causes the HAIM to deform nonhomogeneously, the ionic species will experience differential pressures locally and undergo displacement to give a chemical potential change. In this depolarized state, the change in chemical potential directly corresponded to the change in the electrical potential measured. This electrical potential change, ΔE, can be estimated as in the following equation:

$$\triangle E = \frac{RT}{ZF}\ln\left(\frac{[i]_x}{[i]_{x+\triangle x}}\right) \qquad (1)$$

where $R$ is the gas constant, $T$ is the temperature, $Z$ is the valence of the ion, $F$ is Faraday's constant; $[i]$ represents the concentration of the ith species and the subscripts represent two locations separated by a distance $\Delta x$ within the sample. The detailed mathematical derivation is shown in Supplementary Fig. 11. Clearly, the potential gradient of the HAIM is determined by the value of $[i]_x/[i]_{x+\triangle x}$. Higher contents of ions in the pristine iontronic host can easily be displaced to achieve a significant ion gradient providing a larger potential gradient of −64.5 mV. However, due to diffusion and drifting of free ions in the pristine iontronic host, the resultant potential gradient decreased to −1 mV after a 1 min interval. Moreover, due to the photodimerization crosslinking and IL secretion among primary patterns, the IL content decreased and the ion gradient was reduced, causing the potential gradient between the primary patterns and the unexposed region to decreas to −21.6 mV and decline to −5 mV after 1 min. In particular, the potential gradient between IL-doped PEDOT on primary patterns and the raw PEDOT on secondary wrinkles was stably maintained at −35.7 mV due to the stable ion gradient difference, as shown in Fig. 1f. Thus, vapor polymerization of the asymmetric PEDOT plays a vital role in generating the potentiometric gradient.

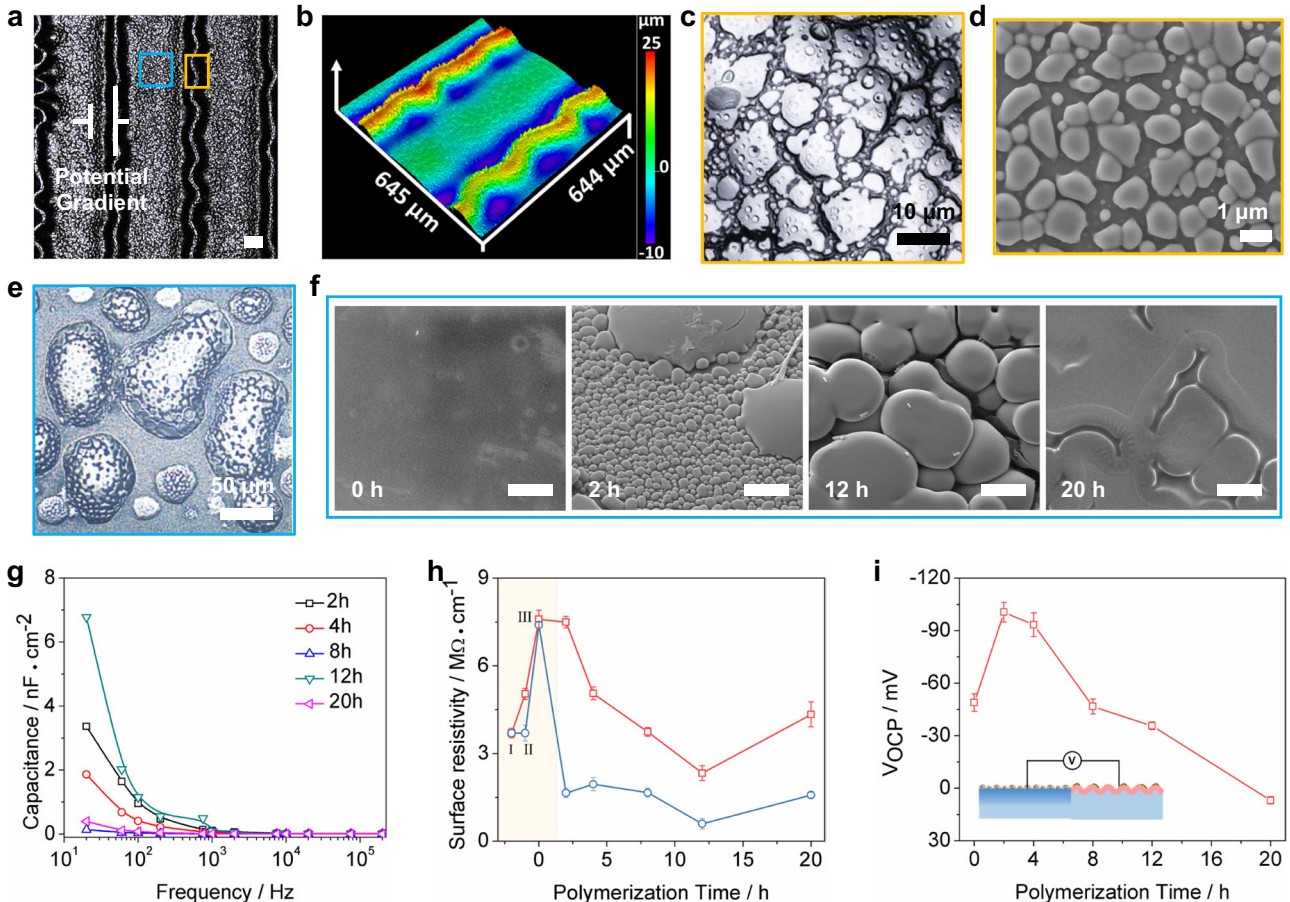

**Fig. 6 | Regulation of the potentiometric gradient. a** Microscopy image of the as-prepared striped HAIM; the blue and yellow frames represent the IL-doped PEDOT @primary patterns and raw PEDOT @ secondary wrinkles, respectively. **b** LSCM images of the HAIM stripes corresponding to **a**. **c** Microscopy images and **d** SEM images of raw PEDOT on secondary wrinkles. **e** Microscopy images of as-prepared IL-doped PEDOT after polymerization for 12 h. **f** SEM images of IL-doped PEDOT with different polymerization times, scale bar: 5 μm. **g** Interfacial capacitance changes of as-prepared striped HAIM with different polymerization times and frequencies. **h** Surface resistivity of as-prepared HAIM stripes (red line and gray-blue line represent the exposed and the unexposed regions, respectively) prepared with different polymerization times; Before polymerization started, I, II, and III represented pristine iontronic host, IL droplets @primary patterns, and IL @primary patterns after absorbing EDOT, respectively. Error bars represent the standard deviations of five independent data. **i** Potentiometric gradients of HAIM at different polymerization times, the inset was the schematic of measurement of potentiometric gradient between IL-doped PEDOT @primary patterns and raw PEDOT @secondary wrinkles. Error bars represent the standard deviations of three independent data.

## Regulation of the potentiometric gradient for HAIM

With the above results in mind, we further demonstrated control of the vapor polymerization time as the key to regulating the potentiometric gradient. To easily integrate the HAIM into an external circuit, an evenly spaced stripe-like HAIM was prepared, which enabled the negative and positive regions to be arranged in a tandem array (Fig. 6a). The LSCM images of the as-prepared HAIM (Fig. 6b) showed a morphology similar to that in Fig. 5b. The micromorphologies of raw PEDOT on secondary wrinkles and the IL-doped PEDOT on primary patterns are shown in Fig. 6c–f. As the polymerization time was increased, spherical IL-doped PEDOT particles gradually joined each other and formed a conductive film on the primary patterns, as shown in Fig. 6f. Additionally, the interface capacitance between the HAIM and the Au-coated interdigital electrodes decreased with longer polymerization times, as shown in Fig. 6g. Notably, the largest interfacial capacitance of the HAIM appeared for a polymerization time of 12 h and decreased slowly with frequency changes. The higher interfacial capacitance indicated that more mobile ions were polarized in the IL-doped PEDOT subjected to an external electric field, which was consistent with the surface resistivity variations in Fig. 6h. In terms of the different surface resistivities for the HAIM, both [I]⁻ and [TFSI]⁻ groups could serve as the counterions of PEDOT during

polymerization processes. Regrading IL-doped PEDOT on primary patterns, the IL would exhibit electrostatic interactions with the main chain of the PEDOT and the iodide ions, resulting in a reduction in the structural integrity of the as-prepared PEDOT and a decrease in the conjugation lengths of π bonds[46]. Therefore, the surface resistivity for IL-doped PEDOT was higher than that of raw PEDOT. In addition, the surface resistivity for raw PEDOT on unexposed regions was obviously lower than that of point I (pristine iontronic host), point II (IL droplets @primary patterns), and point III (IL @primary patterns after absorbing EDOT). Specifically, the lowest surface resistivity for raw PEDOT on secondary wrinkles dropped to 0.58 MΩ•cm⁻¹ when the reaction was carried out for 12 h. Moreover, the surface resistivity for IL-doped PEDOT on primary patterns decreased slowly with polymerization time, and the lowest value was 2.33 MΩ•cm⁻¹. Based on these results, we propose that the surface resistivity was initially dominated by the IL content and continuous production of PEDOT, which enabled the HAIM to exhibit higher conductivity. Furthermore, when the reaction time reached 20 h, the excess iodine vapor was adsorbed on the surface of the HAIM, resulting in lower ionic conduction and a higher surface resistivity for the HAIM.

Because the IL-doped PEDOT was filled with free mobile ions, and the main chain regularity of IL-doped PEDOT was low, the ions in their

original equilibrium state were readily redistributed, and thus, an ion gradient was generated under the action of an external force. In addition, the cations in IL-doped PEDOT more easily migrated than the anions under the action of an external force; therefore, excess anions accumulated at the surface, resulting in a negative potential, as shown in Fig. 6i. However, IL-PEDOT gradually formed a film and became rigid as the polymerization time was increased, resulting in reduced ion migration and a gradient due to the external force, which also reduced the open circuit potential gradient between the asymmetric regions. These results indicated that the HAIM could serve as a force-induced electrical generator, implying potential for application in iontronic power generators.

## Discussion

In conclusion, we have demonstrated a down-top, interlocked patterning strategy for scalable fabrication of hierarchical and asymmetrical iontronic micropatterns through a combination of photodimerization and vapor oxidative polymerization. Local photodimerization crosslinking and subsequent heating produced an IL droplet-covered primary pattern-based microreactor and pristine iontronic host-based secondary wrinkles. The resultant IL-doped PEDOT and raw PEDOT on the HAIM responded to external compression and readily generated potential gradients based on the piezoionic effect. The morphology and electrical performance of the HAIM can be controlled with constrained boundary conditions and polymerization time. It is believed that this exceptional fabrication method, which combines the advantages of top-down and bottom-up approaches, will provide a new choice for on-demand surface patterning of iontronic materials. We envision that the HAIM will open opportunities for a range of applications in pressure sensors, force-induced potential generators, and electrochemical deposition.

## Methods

### Fabrication of IL droplets @primary patterns

In this work, anthracene-functionalized ionic liquid copolymer (denoted as An-PIL) was plasticized with 1-ethyl-3-methylimidazolium bistrifluoromethylsulfonimide lithium (denoted as IL). First, 5 wt.% An-PIL solutions were prepared in DMF, and then IL was added into the solutions according to the molar ratios desired for the An-IL segment to IL, 1:0, 1:0.5, 1:1, 1:2, 1:3, 1:5, and 1:10. Thereafter, the homogeneous iontronic hosts were obtained with two rounds of filtration through 0.22 μm Teflon syringe-driven filter. Subsequently, the above mixed solution was drop coated onto a glass plate or PET film, and then an iontronic film with a thickness of approximately 45 μm was obtained after drying at 90 °C for 20 min. Finally, upon irradiation with ultraviolet (UV) light for 15 min (LED UV lamp, 365 nm, 15 mW/cm$^2$), the desired primary patterns were obtained by using different photomasks. Thereafter, the resulting primary patterns were heated at 85 °C for different times to secrete IL droplets from the surface of the primary patterns (denoted as IL @primary patterns).

### Preparation of secondary wrinkles

A homemade hotplate setup was used for EDOT diffusion into IL droplets @primary patterns. First, IL droplets @primary patterns were stuck to a glass petri dish and then placed upside down to cover 5 μL of EDOT on a hotplate. After heating at 120 °C for 20 min, IL droplets @primary patterns with weight increments of 2.9 mg were removed and placed in a crimp cap vial together with iodine (59.7 mg). The crimp cap vials were heated to 70 °C for 2 h, 4 h, 8 h, 12 h, and 20 h in an oven, to realize PEDOT-coated primary patterns and secondary wrinkles (the thickness of the resultant PEDOT layer was ~20 μm for polymerization time of 12 h).

Details of material design, analysis and instruments can be found in the Supplementary Information.

## Data availability

Data supporting the findings of this study are available within the paper and its Supplementary Information files. All other relevant data that support the findings of this study are available from the corresponding author upon request.

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

## Acknowledgements

This work was supported by the National Key R&D Program of China (2021YFB4001100) and the National Natural Science Foundation of China (52025032 and 52103144).

## Author contributions

X.J. and J.Y. conceived the research and analyzed the results and data; Z.W. carried out the material syntheses and characterizations; T.L. and J.L. took part in some of the material syntheses. Y.C. and X.M. tested the application properties. All authors contributed to the manuscript.

## Competing interests

The authors declare no competing interests.
