## [Peer Review File · Nature Communications]

Photodimerization Induced Hierarchical and Asymmetric Iontronic MicropatternsReviewers' Comments:

Reviewer #1:

Remarks to the Author:

In this work, the authors demonstrated a simple and feasible strategy for hierarchical and asymmetrical iontronic micropatterns based on a combination of photodimerization and vapor oxidative polymerization. The morphology and electrical performance can be modulated through programmed regulation of the internal stress distribution and the local ionic migration. The coordinately controlling of the surface topology and electric/ionic conduction for iontronic materials by tuning the spatiotemporal physical/chemical environments is simple but novel and precious compared to previous patterned surface engineering. This is a very interesting work, and the idea shows high novelty. This work would be a great contribution to the field of the buckled structures, for the aspects of basis and applications. Such a strategy may provide a significant reference for the construction of hierarchical ionic patterns and the application of ions-based modality materials. The results and conclusion are well supported by the experimental data. Overall, this is a very decent piece of work. Thus, I recommend this work can be accepted for publication after revision. But I have some nitpicks for the authors to be considering:

1. The author presented different primary patterns locally covered by ionic liquid droplets. How about the results if the categories of ionic liquid varied? Besides, how about the universality of the IL droplets @primary patterns when adapted to be different microreactor vessels for the synthesis of other substances?
2. In section 2.4, primary patterns and secondary wrinkles have been prepared through photodimerization and vapor polymerization, respectively. What is the difference of the mechanism for self-organized primary patterns and secondary wrinkles? In addition, during the hierarchical pattern generation process, what is the migration mechanism of the ionic liquid in the inner or surface layer for different hierarchical patterns?
3. How about the reversibility of the secretion process of ionic liquids on the primary patterns? What are the factors to determine the secretion of ionic liquid upon primary patterns?
4. What are the advantages of this hierarchical patterned surface compared with another hierarchical self-organized patterns (e.g., *Adv. Mater.*, 2021, 33(39): 2007699; *Adv. Funct. Mater.*, 2021, 31(49): 2106754)?
5. In section 2.4, the author mentioned that the higher modulus of primary patterns than that of the pristine iontronic host, however the modulus values of the hierarchical iontronic micropatterns are not presented in the paper.
6. The thickness of the PEDOT layer seems to be not mentioned.
7. In terms of the obtained hierarchical iontronic micropatterns in this paper, due to the different displacements generated by the movable free ions on the surface or in the interior under external compression, the outcome is the potential difference between the hierarchical patterns. Compared with the patterned ion gel prepared by imprinting or template transfer, what advantages does the hierarchical and asymmetrical iontronic micropattern in this paper have in exhibiting the piezoionic effect? How to maximize the functionalization of the piezoionic effect?

Reviewer #2:

Remarks to the Author:

This is an interesting paper that reports a facile preparation of hierarchical and asymmetrical iontronic micropatterns (HAIM) through the combination of photodimerization and vapor oxidative polymerization. The primary patterns and secondary wrinkles with regulated morphology and ionic electrical performance can be readily realized based on the spatiotemporal features of light, heat, and vapor. Additionally, the application potential of the obtained HAIM as one new scalable iontronic potential generator has been demonstrated. This work may open a new avenue toward patterned iontronic materials in a programmable and functionally adaptive form. I would recommend publication if the following concerns are adequately addressed.

1. Figure 1B shows the fabrication process for HAMI through localized photodimerization and vapor oxidative polymerization. Where is the important heating step?
2. As for the mechanism of the photodimerization-induced primary patterning, it was stated that "Thereafter, primary patterns were generated by localized photodimerization of iontronic host with photomasks and that can be understood as the self-wrinkling appeared at the exposure region due to the gradient photocrosslinking (Figure 1B)". This statement is somewhat ambiguous and detailed discussion is needed. Did the striped patterns in the exposed regions (Figure 1c) belong to surface wrinkling? How to control the striped oriented patterns?
3. "Molar ratio of IL" is used as the name of the abscissa in Figure 2D. Should it be molar content?
4. It can be seen from Figure 4F that the heating-induced secondary wrinkles are only generated on the unexposed region. Why cannot they occur on the exposed region with the gradient structures?
5. Figure 4H shows that the wavelength and amplitude change of secondary wrinkles versus polymerization time is relatively complicated. The corresponding recorded AFM images indicate that the microparticle size of the PEDOT also decreases and the film becomes smooth gradually with the polymerization time, which should be considered for the corresponding discussion of the above relation shown in Figure 4H.
6. Scales bars are missing from Figure S3. Moreover, the enlarged images seem not to correlate well to the framed areas.
7. There are numerous grammatical mistakes throughout the manuscript. For example, "Despite complicated surface patterns can" (page 2), "do not conformal to" (page 2), "that supported by" (page 3), "the resemble structure" (page 4), "and which" (page 4), and "the insert were" (page 10).

Reviewer #3:

Remarks to the Author:

This manuscript by Wang et al. presented a novel fabrication technique to achieve hierarchical and asymmetric micropatterns upon iontronic materials. The fabrication process is induced by the programmed regulation of the internal stress distribution via photodimerization and heating in an anthracene-functionalized polymeric ionic liquid. The experimental data and theoretical models are well established to help understand the key factors that affect the morphology and ionic electrical performance of the resultant HAIM. This facile and robust patterning method possesses some advantages compared with structural engineering techniques and it can generate multiple complex iontronic patterns through a convenient approach. Moreover, this method also enables variation of the mobile ionic species to migrate in different patterned regions and thus exhibits the piezo-ionic effect under external compression. The idea of combining photodimerization and vapor polymerization to achieve iontronic micropatterns is innovative. I recommend the publication of this work in NC. Some minor comments are shown below that need to be further clarified.

1. The localized photodimerization of An-PIL is the key point for the fabrication of the HAIM. The authors provided the modified bilayer model that explains the process in detail and matches the experimental results well. The AFM phase images in section 2.2 and Eq. (6~7) in supporting information suggest lower molar ratios of IL in iontronic hosts are favorable for the secretion of droplets. But it is not clear how the molar ratios of IL affect the photodimerization of An-PIL and also the droplets secretion. The authors need to provide a more in-depth discussion on the degree of photodimerization that affects the interaction between IL with the An-PIL.
2. The authors present multiple primary patterns that are covered by ionic liquid droplets in section 2.3. The density and diameters of the ionic liquid droplets within the primary patterns with different shape boundaries also seem to be quite different. How about the resolutions of the resultant clearly separated primary patterns? What is the role of the boundary conditions of the primary patterns in controlling the density of the ionic liquid droplets? What is the limiting boundary scale to form this specific droplet-covered pattern?
3. It would be better to discuss the universality of this method which could make it more widely applicable, i.e., to secrete different ionic liquids and synthesize different conductive polymers. In

addition, an in-depth discussion on the relationships between ionic electronic performance and vapor polymerization conditions would be helpful, as would an empirical law that could be used to predict the iontronic behavior of the properties/approach for desirable patterned iontronic material.

Reviewer #1: In this work, the authors demonstrated a simple and feasible strategy for hierarchical and asymmetrical iontronic micropatterns based on a combination of photodimerization and vapor oxidative polymerization. The morphology and electrical performance can be modulated through programmed regulation of the internal stress distribution and the local ionic migration. The coordinately controlling of the surface topology and electric/ionic conduction for iontronic materials by tuning the spatiotemporal physical/chemical environments is simple but novel and precious compared to previous patterned surface engineering. This is a very interesting work, and the idea shows high novelty. This work would be a great contribution to the field of the buckled structures, for the aspects of basis and applications. Such a strategy may provide a significant reference for the construction of hierarchical ionic patterns and the application of ions-based modality materials. The results and conclusion are well supported by the experimental data. Overall, this is a very decent piece of work. Thus, I recommend this work can be accepted for publication after revision. But I have some nitpicks for the authors to be considering:

Response: We thank you for the very positive comments. The manuscript has been carefully revised according to the suggestions.

Comment 1: *The author presented different primary patterns locally covered by ionic liquid droplets. How about the results if the categories of ionic liquid varied? Besides, how about the universality of the IL droplets @primary patterns when adapted to be different microreactor vessels for the synthesis of other substances?*

Response to 1: Thank you for your comments. We have tried to use several types of ionic liquids to fabricate IL droplets @ primary patterns. Another kind of room temperature ionic liquid [VMIm][NTf₂] has been applied to prepare the IL droplets @primary patterns according to the method presented in section 4.1. As shown in Figure R1, the [VMIm][NTf₂] droplets can similarly secrete onto the primary patterns. In addition, according to Martin Obst's work (Angew Chem Int Ed, 2021, 60: 25668), the IL droplets can be used as crystallization media for KBr, pentacene, and a variety of pharmaceuticals due to their non-volatility and solvent properties. Moreover, thanks to their monomer absorption properties, IL droplets also enable polymerization from the vapor phase, as demonstrated for polyurea and poly (2-hydroxyethyl methacrylate). The synthesis of other substances through our IL droplets @primary patterns is in progress.

Figure R1 Microscopy images of IL droplets @primary patterns comprise of An-PIL and [VMIIm][NTf₂].

Comment 2: *In section 2,4, primary patterns and secondary wrinkles have been prepared through photodimerization and vapor polymerization, respectively. What is the difference of the mechanism for self-organized primary patterns and secondary wrinkles? In addition, during the hierarchical pattern generation process, what is the migration mechanism of the ionic liquid in the inner or surface layer for different hierarchical patterns?*

Response to 2: Thank you for your comments. Theoretically speaking, both primary patterns and secondary wrinkles were resulted from mechanical instability within the system. As for the generation of the primary pattern, when the incident UV light went through the photomask and exhibited a funnel effect normal to the iontronic host, a gradient crosslinking subsequently formed across the exposed area and a modulus difference appeared between the surface crosslinked layer and the raw substrate. Meanwhile, a distinct expansion coefficient also generated between the surface rigid layer and soft substrate with heating from absorbing ultraviolet light and photocrosslinking. When the UV light removed and the exposed iontronic host became cooled, the exposed iontronic finally minimizes the system energy by releasing the stress in the manner of surface buckling. In addition, due to the direction of system stress release can be better controlled by different photomask systems (Adv. Mater. 2018, 30, 1803463), thereby different ordered primary patterns could fabricate. For the fabrication of secondary wrinkles on the surface of the unexposed area, since the unexposed area comprised of pristine iontronic host, it can be regarded as a softer substrate layer. During the process of vapor phase polymerization, the continuous formation of rigid PEDOT on the surface enables the modulus and thickness of the rigid surface layer to increase, and the expansion degree between the surface layer and the substrate layer is also different under heating, so the stress release will ultimately occur through surface wrinkling.

As for the phase distribution of pristine iontronic host, it was highly miscible and well dispersive with the help of the similar structure and close characters between IL and An-PIL backbones (i.e. the strong polarity of An-IL enables the An-IL-rich domains to be surrounded by IL). However, when photodimerization completed and thus the content of An-PIL decreases, the weak polarity of BA segments drives aggregation to form phase-separated BA-rich domains, and therefore the IL originally interacted with An-PIL has become freely mobile. After heating drives the IL to migrate and coalesce and finally secreted out according to the microphase separation in the phase images (Figure 2G-I). Moreover, the mobile IL may also migrate from the unexposed regions to the exposure region and thus enable the An-IL and BA segment units to become more homogenous after localized photodimerization and heating, as shown in Figure 2I.

Comment 3: *How about the reversibility of the secretion process of ionic liquids on the primary patterns? What are the factors to determine the secretion of ionic liquid upon primary patterns?*

Response to 3: Thank you for your comments. The reversibility of the secretion process of IL were showed in Figure R2. When IL droplets @primary patterns were exposed to 254 nm UV light for 15 min, An-PIL dimer began to decrosslinking and more An-PIL moieties generated

as the primary patterns became flattened. Meanwhile, the IL droplets re-dissolved into iontronic host due to the strong polarity of unit of An-IL that enables the IL to be surrounded by. After heating at 130 °C for 5 min, the IL droplets secrete out again but the number and diameter were less and smaller than that of pristine IL droplets @primary patterns. Therefore, heating may contribute to the entropy increase of IL and An-PIL within the iontronic host. Moreover, due to the limited degree to decrosslinking An-PIL dimer by 254 nm UV light, a faintest IL droplet @primary pattern can also regenerate and which was similar to the mechanisms of pristine IL droplets @primary patterns.

Figure R2. The reversibility of the secretion process of ionic liquids on the primary patterns

Comment 4: *What are the advantages of this hierarchical patterned surface compared with another hierarchical self-organized patterns (e.g., Adv. Mater., 2021, 33(39): 2007699; Adv. Funct. Mater., 2021, 31(49): 2106754)?*

Response to 4: Thank you for your comments. In terms of the construction mechanism of micropatterns, reference 1 (Adv. Mater., 2021, 33, 2007699), reference 2 (Adv. Funct. Mater., 2021, 31, 2106754) and this work have different points. Reference 1 utilizes the dimerization reaction of maleimide in the UV exposed regions to generate a concentration gradient between the exposed area and the unexposed area, therefore driving the diffusion of maleimide from the unexposed to the exposed regions. At the same time, due to the attenuation of UV light normal to the film, a concentration gradient also appeared on the surface and bottom of the exposed area, which will also drive the vertical diffusion of maleimide, and thus the diffusion in the two directions jointly promotes the group growth. Furthermore, the resulting diffusion pattern can remain stable due to the small amount of maleimide involved in the D-A cross-linking reaction. In addition, the micropattern growth in reference 2 was caused by the self-assembly of fluorinated polymers containing anthracene groups under UV light irradiation, which induces the formation of hierarchical gradient crosslinks, resulting in hierarchical self-wrinkling patterns on high-modulus surfaces.

Based on the attenuation of ultraviolet light in the direction perpendicular to the film, this paper presents a hierarchical and asymmetrical iontronic micropatterns through programmed regulation of the internal stress distribution and the local ionic migration among an iontronic host. At the same time, due to the weakening of the interaction force between the cross-linked An-PIL dimer and the freely moving IL, the IL droplet movement intensified and coalescence to form larger droplets under the action of heating, and ultimately secreted onto the surface of the exposed area. Furthermore, based on the solvent properties of IL droplets, they were used

as microreactor vessels in vapor phase polymerization of PEDOT. Therefore, realizing the interfacial stress instability of the bilayer system, and finally forming secondary wrinkles. The method presented here has more potential than reference 1 and reference 2 for adjusting the internal content of components and acting as an externally scalable high-throughput microreactor vessel.

Comment 5: *In section 2.4, the author mentioned that the higher modulus of primary patterns than that of the pristine iontronic host, however the modulus values of the hierarchical iontronic micropatterns are not presented in the paper.*

Response to 5: Thank you for your suggestions. The modulus of the HAIM has been added in Figure S9. The corresponding demonstrations also added in section 2.4 and marked in red.

Comment 6: *The thickness of the PEDOT layer seems to be not mentioned.*

Response to 6: Thank you for your comments. The thickness of the PEDOT layer have added in section 4.2 and marked in red.

Comment 7: *In terms of the obtained hierarchical iontronic micropatterns in this paper, due to the different displacements generated by the movable free ions on the surface or in the interior under external compression, the outcome is the potential difference between the hierarchical patterns. Compared with the patterned ion gel prepared by imprinting or template transfer, what advantages does the hierarchical and asymmetrical iontronic micropattern in this paper have in exhibiting the piezoionic effect? How to maximize the functionalization of the piezoionic effect?*

Response to 7: Thank you for your comments. The HAIM presented in this paper are fabricated through the bottom-up process. Specifically, dimerization of An-PIL causes the IL originally stable in the iontronic host to move more easily and then uses the heating to drive coalescence to form larger IL droplets and lastly secrete onto the surface of the exposed area. This process not only realize the preparation of micropatterns on the surface of the ionotron material but also adjust the ionic migration and the final IL content within the ionotron material at the same time. In addition, due to the difference doping degree between the PEDOT generated on the surface of the exposed region and the unexposed region, a better piezoionic effect was achieved. For direct imprinting or template transfer, it's difficult to simultaneously realize the regulation of gradient internal components and the micropatterns generations, and also construction of hierarchical and asymmetric micropatterns on the surface of iontronic materials. Moreover, by adjusting the density and size of ionic liquid droplets secretion, the adsorption time of EDOT vapor, and the vapor phase polymerization time, the piezoionic effect produced by HAIM can be well controlled, as shown in section 2.6 (Figure 6). When the vapor phase polymerization time was 4 h, the maximum iontronic potential generated can reach -100 mV.

Reviewer #2: This is an interesting paper that reports a facile preparation of hierarchical and asymmetrical iontronic micropatterns (HAIM) through the combination of photodimerization and vapor oxidative polymerization. The primary patterns and secondary wrinkles with regulated morphology and ionic electrical performance can be readily realized based on the spatiotemporal features of light, heat, and vapor. Additionally, the application potential of the obtained HAIM as one new scalable iontronic potential generator has been demonstrated. This work may open a new avenue toward patterned iontronic materials in a programmable and functionally adaptive form. I would recommend publication if the following concerns are adequately addressed.

Response: We thank you for the insightful comments. It inspires us very much and the manuscript has been carefully revised according to your good suggestions.

Comment 1: *Figure 1B shows the fabrication process for HAMI through localized photodimerization and vapor oxidative polymerization. Where is the important heating step?*

Response to 1: Thank you for your comments, and we are sorry for the unclear description of Figure 1B. The corresponding icons that represents the heating step has been added in Fig. 1b. Detailed explanations also have been inserted in section 2.1 and marked in red.

Comment 2: *As for the mechanism of the photodimerization-induced primary patterning, it was stated that “Thereafter, primary patterns were generated by localized photodimerization of iontronic host with photomasks and that can be understood as the self-wrinkling appeared at the exposure region due to the gradient photocrosslinking (Figure 1B)”. This statement is somewhat ambiguous and detailed discussion is needed. Did the striped patterns in the exposed regions (Figure 1c) belong to surface wrinkling? How to control the striped oriented patterns?*

Response to 2: Thank you for your good suggestions. The detailed discussion has been added in section 2.1 and marked in red. The resultant striped patterns in the exposed regions (Fig. 1c) belong to one kind of surface wrinkling. Theoretically speaking, wrinkling usually occurs when the system is in mechanical instability, and this phenomenon is believed to be the result of stress relaxation (Soft Matter, 2012, 8, 9086; 2011, 7, 4490). Although wrinkles are generally sinusoidal, orderly rearrangements may occur to form regular patterns under certain stress conditions. Hou reported that local and long-range 2D ordered patterns can fabricate by local buckling of the gradient crosslinked thin film (Adv Mater 2018, 30, 1803463). The control over the topology is given by laterally patterning out-of-plane gradients in the crosslink density of the film. Therefore, based on the surface wrinkling mechanics and strategy of controlling buckling in 2D, various oriented patterns can be readily fabricated by illuminating iontronic host with the desired patterned photomask.

Comment 3: *“Molar ratio of IL” is used as the name of the abscissa in Figure 2D. Should it be molar content?*

Response to 3: Thank you for your comments, and we are sorry for the improper use of the stoichiometries. The abscissa and the capital of Fig. 2D have been corrected. Besides, the corresponding explanations are also corrected in section 2.2 and marked in red. Thank you for your suggestions again.

Comment 4: *It can be seen from Figure 4F that the heating-induced secondary wrinkles are only generated on the unexposed region. Why cannot they occur on the exposed region with the gradient structures?*

Response to 4: Thank you for your comments. The morphology changes of the exposed area during the fabrication of HAIM are shown in Figure R3 below. When heating the exposure area at 85 °C for 15 min (i.e. aim to fabricate IL droplets @primary patterns), dense IL microdroplets were secreted on the surface, as shown in the following Figure R3a. After absorbing EDOT vapor at high temperatures, the diameter of the droplets becomes larger significantly (the following Figure R3b). Meanwhile, a large amount of EDOT dissolved in the surface of the primary patterns and which resulted in stress relaxation within the primary patterns, therefore the primary patterns turned to be a swollen and flat circle, as shown in Figure R3b below. After vapor phase oxidative polymerization, IL-doping PEDOT was produced on the surface of the primary pattern, and the height of the pattern did not change significantly at this time (the following Figure R3c). Since IL-doping PEDOT deposited onto the primary patterns exhibited a lower modulus relative to that of raw PEDOT generated onto secondary wrinkles (the modulus of the IL-doping PEDOT on the exposed regions after vapor phase polymerization for 12 h was 176 MPa, as shown in Supplementary Fig. 9c), the compressive stress between the upper layer (that is, the IL-doping PEDOT) and the lower layer (that is, the An-PIL dimer generated by photodimerization) didn't reach the critical value, and thus wrinkles can't generate on primary patterns (i.e. exposed regions). On the contrary, the raw PEDOT produced in the unexposed regions after vapor phase polymerization for the same time has a higher modulus (the modulus of the raw PEDOT produce on unexposed regions is 1.2 GPa, as shown in Supplementary Fig. 9d), and the bottom layer of the unexposed regions is the relatively softer pristine iontronic host, therefore a large modulus difference and higher compressive stress between the upper and lower layers of the unexposed regions could generate and to enable secondary wrinkles to form.

Figure R3 The optical microscopy images and the corresponding LSCM images of (a) raw IL droplets @primary, (b) EDOT dissolving in raw IL droplets @primary patterns, and (c) the IL-doping PEDOT @primary patterns.

Comment 5: *Figure 4H shows that the wavelength and amplitude change of secondary wrinkles versus polymerization time is relatively complicated. The corresponding recorded AFM images indicate that the microparticle size of the PEDOT also decreases and the film*

becomes smooth gradually with the polymerization time, which should be considered for the corresponding discussion of the above relation shown in Figure 4H.

Response to 5: Thank you for your comments. The wavelength and amplitude changes of secondary wrinkles with polymerization time have been demonstrated in detail in section 2.4 and marked in red. First, since there were a few PEDOT generated, so they can't join with each other, resulting in flaky PEDOT @secondary wrinkles (polymerization for 2 h). Pristine EDOT and resultant PEDOT coexisted on the surface of the secondary wrinkles at this time. As the polymerization time increased, EDOT gradually polymerized to be PEDOT. The generated PEDOT becomes dense and tends to form a film. Meanwhile, the excess stress also concentrated within bilayers comprises rigid PEDOT and soft secondary wrinkles. At this point, the amplitude and wavelength of the PEDOT @secondary wrinkles also dropped sharply for the dramatic compression stress release via crack formation (polymerization for 12 h in Fig.4g). When vapor oxidative polymerization lasts longer (polymerization for 20 h), PEDOT continues to generate onto cracked PEDOT @secondary wrinkles, and gradually becomes smooth. The wavelength and amplitude of PEDOT @secondary wrinkles and cracks also become larger due to the further accumulation of stress. In a word, the microparticle and thickness morphology of PEDOT could impact the wavelength and amplitude change of secondary wrinkles via changing the stress. Thank you for your suggestions again.

Comment 6: *Scales bars are missing from Figure S3. Moreover, the enlarged images seem not to correlate well to the framed areas.*

Response to 6: Thank you for your suggestions. The scale bars have been added in Figure S3, and the framed areas also have been corrected.

Comment 7: *There are numerous grammatical mistakes throughout the manuscript. For example, "Despite complicated surface patterns can" (page 2), "do not conformal to" (page 2), "that supported by" (page 3), "the resemble structure" (page 4), "and which" (page 4), and "the insert were" (page 10).*

Response to 7: Thank you for your comments. All grammatical mistakes that you mentioned in the manuscript have been corrected carefully. In addition, we have polished the whole manuscript with the help of Spring Nature Author Services (please verified on the SNAS website using the verification code DCB4-361E-5F52-A868-1E5P), Thank you for your suggestions again.

Reviewer #3: This manuscript by Wang et al. presented a novel fabrication technique to achieve hierarchical and asymmetric micropatterns upon iontronic materials. The fabrication process is induced by the programmed regulation of the internal stress distribution via photodimerization and heating in an anthracene-functionalized polymeric ionic liquid. The experimental data and theoretical models are well established to help understand the key factors that affect the morphology and ionic electrical performance of the resultant HAIM. This facile and robust patterning method possesses some advantages compared with structural engineering techniques and it can generate multiple complex iontronic patterns through a convenient approach. Moreover, this method also enables variation of the mobile ionic species to migrate in different patterned regions and thus exhibits the piezo-ionic effect under external compression. The idea of combining photodimerization and vapor polymerization to achieve iontronic micropatterns is innovative. I recommend the publication of this work in NC. Some minor comments are shown below that need to be further clarified.

Response: We thank you for the very positive and precious comments. The manuscript has been carefully revised based on the suggestions.

Comment 1: *The localized photodimerization of An-PIL is the key point for the fabrication of the HAIM. The authors provided the modified bilayer model that explains the process in detail and matches the experimental results well. The AFM phase images in section 2.2 and Eq. (6~7) in supporting information suggest lower molar ratios of IL in iontronic hosts are favorable for the secretion of droplets. But it is not clear how the molar ratios of IL affect the photodimerization of An-PIL and also the droplets secretion. The authors need to provide a more in-depth discussion on the degree of photodimerization that affects the interaction between IL with the An-PIL.*

Response to 1: Thank you for your comments. As for the phase distribution of pristine iontronic host, it was highly miscible and well dispersive with the help of the similar structure and close characters between IL and An-PIL backbones (i.e. the strong polarity of An-IL enables the An-IL-rich domains to be surrounded by IL). However, when photodimerization was completed and thus the content of An-PIL decreases, the weak polarity of BA segments drives aggregation to form phase-separated BA-rich domains, and meanwhile, the IL originally interacted with An-PIL and became freely mobile. After heating drives the IL to migrate and coalesce and finally secreted out according to the microphase separation in the phase images (Fig 2g-i). Moreover, the mobile IL may also migrate from the unexposed regions to the exposure region and thus enable the An-IL and BA segment units to become more homogenous after localized photodimerization and heating, as shown in Fig. 2i.

The in-depth discussion on the degree of photodimerization that affects the interaction between IL with the An-PIL have been added in section 2.2 and marked in red.

Comment 2: *The authors present multiple primary patterns that are covered by ionic liquid droplets in section 2.3. The density and diameters of the ionic liquid droplets within the primary patterns with different shape boundaries also seem to be quite different. How about the resolutions of the resultant clearly separated primary patterns? What is the role of the boundary*

conditions of the primary patterns in controlling the density of the ionic liquid droplets? What is the limiting boundary scale to form this specific droplet-covered pattern?

Response to 2: Thank you for your comments. The morphology of IL droplets @primary patterns where we selectively illuminated the iontronic host with masks of different diameters and then heated at 85 °C for 15 min are shown in following Figure R4. As the diameter of the circles continues to decrease, the resulting primary patterns become more obvious. Therefore, it can be considered that as the diameter of the boundary decreases, the stress is relatively more concentrated, and thus higher micropatterns are produced. In addition, with the decrease of the mask diameter, the boundary of the ionic liquid droplets within the exposed area also gradually disappeared from initial clearness, and the density of the ionic liquid droplets also decreases. In particular, when the size of the reticle is 50 μm, the secreted ionic liquids start to coalesce with each other. Therefore, it can be considered that the minimum resolution for forming IL droplets @primary patterns can reach 50 μm.

Figure R4 The microscopy images of IL droplets @primary patterns against different diameters of photomask.

Comment 3: *It would be better to discuss the universality of this method which could make it more widely applicable, i.e., to secrete different ionic liquids and synthesize different conductive polymers. In addition, an in-depth discussion on the relationships between ionic electronic performance and vapor polymerization conditions would be helpful, as would an empirical law that could be used to predict the iontronic behavior of the properties/approach for desirable patterned iontronic material.*

Response to 3: Thank you for your comments. Another kind of room temperature ionic liquid [VMIm][NTf₂] has been used to prepare the IL droplets @primary patterns according to the method presented in section 4.1. As you can see from Figure R5, the [VMIm][NTf₂] droplets similarly secreted onto the primary patterns. In addition, according to Martin Obst's work (Angew Chem Int Ed, 2021, 60: 25668), the IL droplets can be used as crystallization media for KBr, pentacene, and a variety of pharmaceuticals due to their non-volatility and solvent properties. Moreover, thanks to their monomer absorption properties, IL droplets also enable polymerization from the vapor phase, as demonstrated for polyurea and poly (2-hydroxyethyl

methacrylate). The synthesis of other substances through our IL droplets @primary patterns is in progress.

Moreover, an in-depth discussion on the relationships between ionic electronic performance and vapor polymerization conditions have been added in section 2.5 & 2.6 and marked in red.

Figure R5 Microscopy images of IL droplets @primary patterns comprise of An-PIL and [VMIm][NTf₂].

Reviewers' Comments:

Reviewer #1:

Remarks to the Author:

now the manuscript can be accepted.

Reviewer #2:

Remarks to the Author:

The authors have properly addressed my previous concerns and the quality of this work has improved. I would be happy to recommend acceptance of this manuscript for publication.

Reviewer #3:

Remarks to the Author:

In the revised version, the main concerns have been well addressed. I recommend the acceptance of the manuscript for the publication by Nat. Commun.

Reviewer #1 (Remarks to the Author): now the manuscript can be accepted.

Response: Thank you again for your comments.

Reviewer #2 (Remarks to the Author): The authors have properly addressed my previous concerns and the quality of this work has improved. I would be happy to recommend acceptance of this manuscript for publication.

Response: Thank you for your good suggestions and comments to strengthen the manuscript.

Reviewer #3 (Remarks to the Author): In the revised version, the main concerns have been well addressed. I recommend the acceptance of the manuscript for the publication by Nat. Commun.

Response: Thank you for your positive comments.